# Cumulus Cells Accelerate Postovulatory Oocyte Aging through IL1–IL1R1 Interaction in Mice

**DOI:** 10.3390/ijms24043530

**Published:** 2023-02-09

**Authors:** Xin Wen, Qi Yang, Dui Sun, Zhao-Yu Jiang, Teng Wang, Hao-Ran Liu, Zhe Han, Lu Wang, Cheng-Guang Liang

**Affiliations:** State Key Laboratory of Reproductive Regulation & Breeding of Grassland Livestock, School of Life Sciences, Inner Mongolia University, Hohhot 010070, China

**Keywords:** cumulus cells, oocytes, POA, IL1, IL1R1

## Abstract

The oocytes of female mammals will undergo aging after ovulation, also known as postovulatory oocyte aging (POA). Until now, the mechanisms of POA have not been fully understood. Although studies have shown that cumulus cells accelerate POA over time, the exact relationship between the two is still unclear. In the study, by employing the methods of mouse cumulus cells and oocytes transcriptome sequencing and experimental verification, we revealed the unique characteristics of cumulus cells and oocytes through ligand–receptor interactions. The results indicate that cumulus cells activated NF-κB signaling in oocytes through the IL1–IL1R1 interaction. Furthermore, it promoted mitochondrial dysfunction, excessive ROS accumulation, and increased early apoptosis, ultimately leading to a decline in the oocyte quality and the appearance of POA. Our results indicate that cumulus cells have a role in accelerating POA, and this result lays a foundation for an in-depth understanding of the molecular mechanism of POA. Moreover, it provides clues for exploring the relationship between cumulus cells and oocytes.

## 1. Introduction

Mature ovulated oocytes that are not fertilized in time will experience aging in a time-dependent manner, also known as postovulatory oocyte aging (POA) [1,2,3]. POA occurs in many research and clinical applications, including in vitro oocyte manipulation or pre-fertilization oocyte culture [4]. These in vitro cultures may cause decline in the oocyte quality and accelerate oocyte aging, resulting in various cellular and molecular changes [5,6,7], including mitochondrial dysfunction, a significant increase in reactive oxygen species (ROS), and early apoptosis [8,9,10,11]. Therefore, researchers have made great efforts to reveal the mechanism of POA. However, the detailed molecular mechanism remains unclear.

Usually, cumulus–oocyte complexes (COCs) are formed from mature oocytes and peripheral cumulus cells [12,13]. Cumulus cells promote oocyte maturation, ovulation, and fertilization [14,15]. Although a few studies have shown that the existence of intact cumulus cells in vitro can protect oocytes from a decline in quality caused by increased oxidative stress [16], several studies have confirmed that cumulus cells may promote oocyte aging and reduce the quality of oocytes [17,18,19]. In addition, FasL released by apoptotic cumulus cells binds to the Fas receptor on oocytes, increasing the sensitivity of oocytes to activating stimuli [19]. In brief, oocytes from COCs or co-cultured with cumulus cells undergo aging faster than denuded oocytes [17]. However, whether other mechanisms play roles in cumulus cells accelerating POA is still unknown.

NF-κB signaling is essential in regulating cellular responses [20,21]. Extracellular stimuli that activate NF-κB signaling are cytokines, which are initially small-molecule proteins synthesized and are secreted by immune cells [22]. The binding of cytokines with their receptors initiates complex interactions between molecules, leading to changes in the downstream signaling pathway [23]. Proinflammatory cytokines stimulate the release of nuclear factor-κB-p65 (RELA) and nuclear factor-κB subunit 1 (NFKB1) heterodimers. Activation of RELA and NFKB1 heterodimers further induces target gene expression and regulates intracellular homeostasis after translation [24]. Interleukin 1 (IL1) is one of the critical proinflammatory cytokines, which can be divided into IL1A and IL1B [25,26]. Freshly isolated cumulus cells have been shown to express IL1A and IL1B [27]. IL1A and IL1B can lead to cell apoptosis. In the PCOS model, excessive hyperandrogenism stimulates cumulus cells secreting IL1A and IL1B, which in turn promotes oocyte apoptosis [28]. IL1A and IL1B usually bind to the receptor on the cell surface and activate the downstream signaling pathway. The IL1 signal is transmitted through the type 1 (IL1R1) but not the type 2 (IL1R2) IL1 receptor, both of which exist in many cells [29,30]. There is a large amount of IL1RI in the cytoplasm and plasma membrane of mouse oocytes [29,31]. These results indicate the presence of the ligand IL1 in cumulus cells and its corresponding receptor in oocytes [30]. It was shown that preimplantation embryos establish a relationship with maternal endometrium via IL1–IL1R1 interaction [32]. This indicates that IL1–IL1R1 interaction plays an essential role in human or mouse reproduction [33]. However, whether IL1–IL1R1 interaction participates in POA accelerated by cumulus cells needs to be explored and confirmed.

In this study, we used transcriptomics analysis and experimental validation to study the mechanism of POA acceleration by cumulus cells. Addressing this question will help us understand the detailed mechanism of POA and provide strategies for researching germ cell aging.

## 2. Results

### 2.1. POA Induces Preimplantation Embryo Development Failure

We collected metaphase II (MII) stage COCs. The COCs were randomly divided into several groups: COCs without any additional culture were defined as the control group; COCs with pre-culture in vitro for a specific time were defined as the aging groups. Then, we collected oocytes and cumulus cells separately for morphology, transcriptome, and other analysis. The schematic diagram is illustrated in Figure 1A. To determine the suitable pre-culture time for the generation of the POA model, we tested four durations (6, 12, 18, and 24 h) for oocytes’ in vitro aging. An in vitro fertilization (IVF) experiment was conducted on the oocytes to assess the potential of preimplantation embryo development. No significant difference was observed in the percentage of oocytes cultured for 6 h in vitro and fresh oocytes in each embryo development stage (*p* > 0.05). However, when prolonging the pre-culture time to 12 h, a significant decrease in embryo development was observed compared to that in the control group (*p* < 0.001). Less than 20% of oocytes could develop into blastocysts after fertilization. Moreover, when prolonging the pre-culture time to 18 or 24 h, less than 4% of oocytes could develop into blastocysts (*p* < 0.001). The prolonged culture time of 18 or 24 h induced severe developmental defects. Therefore, we selected 12 h as the suitable pre-culture time for setting up the model of POA (Figure 1B). As shown in Figure 1C, compared with the control group, the aging group had more degenerated embryos. These results indicated that POA significantly impairs preimplantation embryo development.

### 2.2. Various Abnormalities of Oocytes Were Observed in POA

Next, we performed an evaluation of mitochondria distribution inside oocytes. As displayed in Figure 2A, two types of mitochondria distribution were observed in oocytes: aggregated and evenly distributed. The percentage of each type of distribution was analyzed. In the control group, 80.0% of oocytes exhibited an even distribution, while in the aging group, only 37.7% exhibited an even distribution (*p* < 10^−4^, Figure 2B). Then, the production of ROS in oocytes was measured. Confocal images showed that a large proportion of oocytes exhibited strong green fluorescence in the aging group, indicating excessive accumulation of ROS inside oocytes. Conversely, only a small proportion of control oocytes exhibited solid green fluorescence (Figure 2C). The percentage of green fluorescent-positive oocytes significantly differed between the control and the aging groups (*p* < 10^−4^, Figure 2D). Representative images of the early apoptosis of oocytes detected using Annexin-V were shown in Figure 2E. Green fluorescence in the control group can be observed only on the degenerated polar body, but not on the membrane in normal oocytes. However, a green fluorescent circle was detected on the cell membrane when the oocyte underwent early apoptosis. A significant difference was obtained for the percentage of early apoptosis between the control and aging group (*p* < 10^−4^, Figure 2F). These results showed that abnormal mitochondrial distribution, ROS accumulation, and early apoptosis were observed in oocytes with postovulatory aging, which induced the declined oocyte quality.

### 2.3. Changes of Aging-Related Transcription Level in Oocytes

To further illustrate the oocytes’ gene expression change during POA, Smart-seq 2 was performed on the oocytes sample. The principal component analysis (PCA) results showed that the oocytes from the control and aging groups were in different sets (PC1 was 67.5%, PC2 was 26.3%). The two treatments had significant differentiation and good repeatability (Figure 3A). Venn diagrams of expressed gene sets showed a total of 9300 gene sequences in the control oocytes and 8752 genes in the aging oocytes. Among them, 8312 genes were shared by the control and aging groups. In addition, 988 genes were specific to the control group and 440 genes were specific to the aging group (Figure 3B). Next, gene set enrichment analysis (GSEA) was implemented to find differential pathways concerning the expressed genes in the control and aging groups. The top ten GSEA pathways related to the aging group included cytokine–cytokine receptor interaction, cell adhesion molecules, ECM–receptor interaction, axon guidance, regulation of the actin cytoskeleton, melanogenesis, MAPK signaling pathway, pathways in cancer, Parkinson’s disease, and oxidative phosphorylation signaling pathways (Figure 3C). We aimed to identify aging-related gene expression changes in the GSEA pathway to establish the core regulatory factors and the target gene transcription regulation network. The core hubs of aging-related genes in oocytes, included *Il1r1*, *Notch1*, *Map3k7*, *Nfkb1*, *Map3k1*, *Tab2*, *Cxcr3*, *Egfr*, *Tnfsf11*, and *Eif2ak3* (Figure 3D). A volcanic plot of the top ten hub gene expression indicates that the expression of these genes in the control group was upregulated, while the expression of these genes in the aging group was downregulated (Figure 3E). These results suggest that oocytes undergo variation in aging-related transcription levels.

### 2.4. Changes to Aging-Related Transcription Levels in Cumulus Cells

RNA-seq of cumulus cells surrounding the oocytes was performed to detect the changes in cumulus cells during in vitro culture. PCA indicated that the cumulus cells from the control and aging groups were in different sets (PC1 was 78.7%, PC2 was 16.1%). The two treatments had significant differentiation and good repeatability (Figure 4A). Venn diagrams of expressed gene sets showed that 11,275 genes were sequenced in the control cumulus cells and 10,941 genes were sequenced in the aging cumulus cells. Among them, 10,573 genes were shared by control and aging groups. In addition, 702 genes were specific to the control group and 368 genes were specific to the aging group (Figure 4B). We found that the top ten GSEA pathways related to the aging group included arrhythmogenic right ventricular cardiomyopathy, cell cycle, dilated cardiomyopathy, hypertrophic cardiomyopathy, ECM–receptor interaction, focal adhesion, axon guidance, arginine and proline metabolism, cytokine–cytokine receptor interaction, and aminoacyl-tRNA biosynthesis signaling (Figure 4C). We aimed to identify aging-related gene expression changes in the GSEA pathway and establish the core regulatory factors and their target gene transcription regulation network. This analysis revealed that the core hubs of aging-related genes in cumulus cells included *Il6*, *Il1b*, *Cxcl5*, *Cxcl2*, *Cxcl12*, *Cxcl10*, *Cxcl1*, *Ccl4*, *Ccl3*, and *Ccl22* (Figure 4D). The volcanic plot of the top ten hub gene’s expression was downregulated and upregulated for control cumulus cells and aging cumulus cells, respectively (Figure 4E). These results indicate that aging-related transcription levels in cumulus cells changed.

### 2.5. Il1–Il1r1 Participate in Interactions between the Cumulus Cells and Oocytes

Based on the above transcriptome results, the interaction pathways between oocytes and cumulus cells were analyzed jointly. Combined analysis of PCA in cumulus cells and oocytes showed that different groups were in different positions, and all had reliable intergroup repeatability (Figure 5A). After comparing the top ten GSEA pathways in each group, we found that cytokine–cytokine receptor interaction, axon guidance, and ECM–receptor interaction pathways co-existed in both cumulus cells and oocytes. In GSEA analysis, *p*-value < 0.05, q-value < 0.25, and |NES| > 1 indicate that the results are meaningful. We focused on the cytokine–cytokine receptor interaction because the gene set was expressed at a high level in the cumulus cells and a low level in oocytes (cumulus cells: *p*-value = 0.001, q-value = 0.037, NES = 1.794; oocytes: *p*-value = 0.001, q-value = 0.036, NES = −2.017) (Figure 5B). Further, the spin plot confirmed the related genes’ expression profile in the target pathway; all genes were lowly expressed in control cumulus cells and highly expressed in aging cumulus cells. On the contrary, the genes were highly expressed in control oocytes and lowly expressed in aging oocytes (Figure 5C). This pathway found six ligand–receptor pairs between cumulus cells and oocytes, including *Cxcl10–Cxcr3*, *Il1a–Il1r1*, *Il1b–Il1r1*, *Il1a–Il1r2*, *Il1b–Il1r2*, and *Eda–Edar* (Figure 5D). Next, we assessed the expression levels of each pair of ligand–receptor. The results showed that *Cxcl10–Cxcr3* and *Eda–Edar* had no significant difference between groups (*p* > 0.05). Notably, pairs of *Il1a–Il1r1*, *Il1b–Il1r1*, *Il1a–Il1r2*, and *Il1b–Il1r2* were expressed at a low level in the control group and a high level in the aging group (*p* < 10^−3^) (Figure 5E). Considering that *Il1r2* does not take part in the activation of subsequent signaling pathways, we focused on the ligand–receptor relationship of *Il1a–Il1r1* and *Il1b–Il1r1* (Figure 5F). It is conjectured that *Il1–Il1r1* participates in interactions between the cumulus cells and oocytes.

### 2.6. NF-κB Signaling Is Activated by the IL1–IL1R1 Interaction

To confirm the transcriptome results’ accuracy, mRNA expression of *Il1* in cumulus cells was detected by RT-qPCR. The mRNA level of ligands *Il1a* and *Il1b* was higher in the aging cumulus cells than in the control cumulus cells (*p* < 10^−4^, Figure 6A,B). Conversely, the mRNA level of receptor *Il1r1* in the aging oocytes was lower than that of the control oocytes (*p* < 10^−4^, Figure 6C). Western blot showed that the ligands IL1A and IL1B in aging cumulus cells were lower (*p* < 0.05, Figure 6D–G). Similarly, the receptor IL1R1 in aging oocytes was lower (*p* < 0.05, Figure 6H,I). Using protein interaction network analysis, we found that ligands IL1A, IL1B, and receptor IL1R1 were closely related to RELA and NFKB1, which are core proteins in NF-κB signaling (Figure 6J). Subsequently, significant statistical differences were found in the protein expression levels of RELA and NFKB1 in aging and control oocytes. The RELA and NFKB1 protein expression level of aging oocytes was higher (*p* < 0.01, Figure 6K–N), indicating that NF-κB signaling was activated in oocytes with postovulatory aging. The above data illustrate that during POA, IL1A and IL1B in cumulus were synthesized and released from the cumulus cells, binding to receptor IL1R1 in oocytes, then activating NF-κB signaling in oocytes to initiate a variety of abnormalities in oocytes.

## 3. Discussion

This study revealed a mechanism whereby cumulus cells accelerated POA through joint omics and verification experiments on mice cumulus cells and oocytes. Namely, cumulus cells accelerated the occurrence of POA through IL1–IL1R1 in cytokine–cytokine receptor interaction. Therefore, many abnormal phenomena and low oocyte quality were observed in oocytes with postovulatory aging. This work provides information for understanding the new mechanism behind POA and clues for further studying the interaction between cumulus cells and oocytes in clinical research.

After prolonged in vitro culture for oocytes, many abnormalities can be found, including abnormal aggregated mitochondrial distribution, abnormal increased ROS level, early apoptosis occurrence, and preimplantation embryo development failure [7,34]. It has been found that more than 40% of aging oocytes exhibit disordered aggregated mitochondrial distribution patterns in the cytoplasm [35]. The existence of cumulus cells can enable oocytes cultured in vitro to accumulate a large amount of ROS [36]. It can also induce cell apoptosis and thus reduce oocyte quality [37]. The above abnormality in oocytes with postovulatory aging will eventually affect the embryonic development potential after oocyte fertilization [38]. These previous results were consistent with our current study, wherein oocyte quality abnormality and preimplantation embryo development failure were observed during oocytes’ in vitro culture.

Cumulus cells have been implicated in regulating oocyte development and oocyte–sperm interaction [39,40,41]. It has been found that some ions and small molecules can be transferred between cumulus cells and oocytes, facilitating signal communication between these two types of cells [42]. Cumulus cells accelerate the aging of mouse oocytes by secreting soluble and heat-sensitive paracrine factors [43]. It was confirmed that the factor causing this might be Fasl or TNF-α [18,19]. The development of omics provides a basis for an in-depth understanding of molecular changes in cells [44]. Transcriptome analysis of cumulus cells and oocytes revealed molecular pathways involved in reproductive aging physiology [45], such as cases where the combined analysis of cumulus cells and oocyte transcriptome sequencing revealed age-related effects influenced by life stage and calorie restriction [46]. Our study used Smart-seq 2 and RNA-seq for transcriptome sequencing in oocytes and cumulus cells, respectively. According to the relevant reports of Mishina’s use of hub genes [46], we found the top ten GSEA pathways and top ten hub genes of oocytes and cumulus cells. A combined analysis of the omics data showed that the cytokine–cytokine receptor interaction in the top ten GSEA pathways displayed abnormalities in both aging cells. This pathway is one of the key intercellular pathways that ensure normal cell development [47]. The results of cumulus cell transcriptome analysis from patients with endometriosis showed that the abnormality of cytokine–cytokine receptor interactions affected oocyte quality [48]. Meanwhile, transcriptome data demonstrated that this pathway plays an essential role in oocyte maturation and fertilization [49]. Further analysis of the top ten hub genes in the cytokine–cytokine receptor interaction showed various ligand–receptor relationships. Ligand–receptor pairs can transmit signals between cumulus cells and oocytes. The complex gene regulatory circuit between oocytes and surrounding cumulus cells by transcriptome data might be related to oocyte quality and embryonic development potential [50]. This is similar to the findings in our study of the effect of ligand–receptor interactions on oocyte quality through the combined analysis of cumulus and oocyte transcriptome.

In this study, the mRNA and protein expression results of ligands IL1A and IL1B in cumulus cells and receptors IL1R1 in oocytes were consistent with the transcriptome data. The mRNA levels of ligands *Il1a* and *Il1b* were increased and the protein levels were decreased in cumulus cells from the aging group. Therefore, we propose that cumulus cells would continuously synthesize mRNA after external stimulation during in vitro culture [27], as shown by increased mRNA levels of *Il1a* and *Il1b*, which generate more protein [51]. However, we found that the IL1A and IL1B protein expression levels decreased. It may be due to ligands’ protein being secreted extracellularly after enzymolysis [52,53]. In addition, ubiquitin–proteasome is also a common mode of endogenous protein degradation [54]. For example, IL1A and IL1B proteins are polyubiquitinated and degraded by the proteasome in mouse bone marrow-derived dendritic cells [55]. Therefore, we speculate that the endogenous proteins IL1A and IL1B in cumulus cells may also undergo ubiquitin–proteasome action, leading to reduced protein expression. This explains that mRNA and protein expression levels of IL1A and IL1B are inconsistent in aging cumulus cells. IL1R1′s mRNA and protein expression are reduced in aging oocytes because mammalian oocytes undergo mRNA decay, resulting in decreased mRNA expression [56,57]. The receptor proteins are combined with ligands and are then internalized by cells. After internalization, some receptors are degraded or reused by the cell membrane, which can reduce the number of receptor proteins [58,59]. Although the expression of receptor IL1R1 protein decreased in oocytes, the downstream signaling pathway is activated after ligand–receptor binding. Cumulus cells were found to activate NADPH oxidases in oocytes through the ligand–receptor (Fas–Fasl) system to accelerate POA [19]. Therefore, the ligand–receptor interaction between the cumulus cells and oocytes plays an important role in accelerating POA. In addition to the Fas–FasL system, there may be other alternative ligand–receptor systems for cumulus cells to accelerate POA.

According to the protein–protein interaction network analysis, IL1–IL1R1 is related to RELA and NFKB1 proteins, which are the critical proteins in NF-κB signaling [60]. This study verified the increased expression of RELA and NFKB1 proteins in aging oocytes. Expression of RELA was increased in the liver cells in *Il1r1* conditional deletion mice [61]. The results in the liver injury model were consistent with those in this study, in which a decreased IL1R1 expression was associated with an increased RELA expression. This suggests that the relationship between IL1R1 and RELA in somatic cells may also exist in germ cells. Although there is no direct evidence that the decrease in IL1R1 protein expression is related to the increase in NFKB1 protein expression, RELA usually forms heterodimers with NFKB1 to exert biological effects and RELA protein expression is always positively correlated with NFKB1 protein expression [62,63]. Therefore, the decrease in IL1R1 protein and the increase in RELA and NFKB1 protein expression found in this study are reasonable. In addition, the increased expression of RELA and NFKB1 indicates the activation of NF-κB signaling [64,65,66]. Activating NF-κB signaling leads to biochemical changes, including mitochondrial dysfunction and redox state disorders in skeletal muscle cells [67]. Similarly, mitochondrial dysfunction and elevated ROS levels have been observed after the IL1 activated NF-κB signaling in ataxic telangiectasia fibroblasts [68]. This is consistent with our results that cumulus cells interact with oocytes through the IL1–IL1R1 system in cytokine–cytokine receptor interactions and activates NF-κB signaling in oocytes by increasing the expression of RELA and NFKB1, leading to mitochondrial dysfunction, abnormal ROS accumulation, and increased early apoptosis. In sum, our results illustrate the decline in oocyte quality during POA and the potential role of cumulus cells in this process. Studying the relationship between oocytes and cumulus cells in vitro can provide a theoretical basis for revealing the detailed mechanism of POA. However, the detailed relationship between them still needs further investigation.

## 4. Materials and Methods

### 4.1. Animals and Experimental Design

The Laboratory Animal Management Committee of Inner Mongolia University has approved all animal experiments performed in this study (authorization number: SYXK2020-0006). This study used 200 ICR female mice (8 weeks old), including 15 mice in the transcriptome analysis and 185 mice in other experiments. In addition, three male ICR mice (16 weeks old) were used for sperm collection during IVF. The mice were purchased from SBF Biotechnology Co., Ltd. (Beijing, China) and raised in an SPF-class housing laboratory at Inner Mongolia University. Female mice were killed by cervical dislocation, and COCs from the ampulla of the fallopian tube were collected. All the COCs were pooled and incubated in a Chatot–Ziomek–Bavister (CZB) medium for 10 min. The oocytes were randomly divided into the control group and the aging group. Oocytes in the control group were immediately used for subsequent treatments without prolonged in vitro culture, while the oocytes in the aging group were subjected to extended in vitro culture before use. The cumulus cells and oocytes were stripped, and these two types of cells were separately collected for morphological detection, transcriptome analysis, and other experiments (Figure 1A).

### 4.2. Cumulus Cells and Oocytes Collection

For superovulation, 5 IU pregnant mare serum gonadotropin (PMSG, San Sheng, Ningbo, China) was injected subcutaneously into female ICR mice, 5IU human chorionic gonadotropin (hCG, San Sheng) was intraperitoneally injected into the same female ICR mice 48 h later, and COCs were collected from the oviduct 14 h later. After the corresponding treatment, the COCs were rinsed in the M2 medium containing 0.3 mg/mL hyaluronidase, and cumulus cells and oocytes were stripped, respectively. After PBS washing, the oocytes were collected. In addition, cumulus cells were collected after centrifuging at 1500 rpm for 3 min.

### 4.3. IVF Experiment and Embryo Culture

IVF was performed as previously described [69]. One sexually mature male mouse was used in each IVF repeat. A total of 3 repeats were conducted. All male ICR mice have been proven capable of fertilization through mating experiments. For sperm capacitation, male mice were killed by cervical dislocation. Sperm collected from the tail of both epididymides were mixed, and the sperm was suspended in 200 μL Tyrode (T6) medium supplemented with 10 mg/mL Bovine Serum Albumin (BSA) for 1.5 h. For fertilization, the obtained COCs were placed in 200 μL T6 medium supplemented with 20 mg/mL BSA, and 10 μL of capacitated sperm was added (final concentration is 1 × 10^6^/mL). After COCs and capacitated sperm were co-cultured for 6 h, the fertilized oocytes were transferred and cultured in CZB medium under a humidified atmosphere of 5% CO_2_ at 37 °C. Embryos were transferred to a CZB medium supplemented with 5.5 mmol/L glucose when they reached the 4-cell stage. The percentage of embryos reaching the 2-cell stage was used for fertilization evaluation. The early embryonic developmental potential ratios at each stage were recorded at the corresponding time.

### 4.4. Mitochondria Distribution Detection

Oocytes were fixed in PBS containing 4% (*w*/*v*) paraformaldehyde (PFA) for 20 min and incubated in the CZB medium containing Mito-Tracker Green (C1048, Beyotime, Shanghai, China) for 25 min to detect the oocytes’ mitochondria distribution. Afterward, oocytes were photographed on the cell imaging dishes with a confocal laser scanning microscope (CLSM, Nikon, A1R, Tokyo, Japan). Image J (http://rsbweb.nih.gov/ij/, accessed on 1 September 2022) software (version 1.53t) is utilized to measure oocytes’ mitochondrial fluorescence intensity.

### 4.5. ROS Assay

The ROS production inside oocytes was detected by incubating oocytes in the CZB medium containing DCFH-DA (S0033S, Beyotime, Haimen, China) at 37 °C and 5% CO_2_ for 25 min. Oocytes were transferred to cell imaging dishes and photographed with CLSM. Image J software (version 1.53t) was used to measure ROS fluorescence intensity in oocytes.

### 4.6. Early Apoptosis Detection

Oocytes were stained for 25 min with an Annexin-V-FITC (Vazyme, Nanjing, China) at 37 °C and 5% CO_2_ to detect the occurrence of early apoptosis. Oocytes were transferred to cell imaging dishes and photographed with CLSM. Fluorescent signals on the oocyte membrane were specified as early apoptosis. Image J software (version 1.53t) was utilized to measure and record oocytes’ early apoptosis.

### 4.7. Transcriptome Sequencing of Cumulus Cells and Oocytes

This study mainly explored the interaction between oocytes and cumulus cells. Considering the small number and preciousness of oocytes, micro-cell transcriptome sequencing (Smart-seq 2) methods were used to detect oocytes. However, the cumulus cells surrounding the oocyte are more extensive, so the cumulus cells were subjected to conventional transcriptome sequencing (RNA-seq) methods. Forty oocytes and corresponding cumulus cells were harvested from the control and aging groups, respectively. The oocytes and cumulus cells were separately collected as samples for one sequencing. A total of three separate biological replicates sequencing were performed. In other words, in the transcriptome sequencing experiment, a total of 120 oocytes and corresponding cumulus cells were used in the control group, and a total of 120 oocytes and corresponding cumulus cells were used in the aging group. The effective concentration of the cDNA library (>2 nM) was quantified accurately after each sample’s RNA purity, concentration, and integrity detection. Different libraries were pooled according to the target data volume and sequenced on the Illumina novaseq 6000 platforms with 150 bp paired-end reads. During analysis, clean reads in the FASTQ format were obtained after filtering all generated raw reads. After the alignment analysis, String Tie (version 1.3.4d) used fragments per kilobase of transcript per million fragments mapped (FPKM) to measure the transcript level and was used to assemble and quantify alignment reading. The cutoff threshold for the significance specified by the edge R was log2 (fold change) ≥ 1.5 and *p*-value ≤ 0.01. The GSEA and the KEGG pathway were utilized to determine the pathway of significant enrichment in cumulus cells and oocytes. Other analyses were performed on the BMK Cloud Platform (www.biocloud.net accessed on 29 January 2022).The gene count values of cumulus cells and oocytes are included in the Appendix A.

### 4.8. Cell Communication Inference

CellTalkDB (a database created from a protein–protein interaction network in the STRING database) containing mouse ligand–receptor pairs supported by the literature was used to identify crosstalk signals of transcriptome sequencing data from oocytes and cumulus cells.

### 4.9. Real-Time Quantitative PCR

Total RNA in cumulus cells and oocytes was extracted and inverted into cDNA using a PicoPure RNA Extraction Kit (kit0204, Thermo Fisher Scientific, Waltham, MA, USA) and PrimeScript RT Kit (RR047A, Takara, Dalian, China), respectively. A TB Green kit (RR420A, TaKaRa) and LightCycle 480 system were used to perform real-time RT-qPCR experiments to obtain the cycle threshold (Ct) values (2^−ΔΔCt^) formula to calculate the relative ratio of the sample’s transcription level to the control group to obtain folding changes. Primer sequences included *Il1a* (F: 5′-CCATCCAACCCAGATCAGCA-3′; R: 5′-GTTTCTGGCAACTCCTTCAGC-3′), *Il1b* (F: 5′-GCTGAAAGCTCTCCACCTCA-3′; R: 5′-AGGCCACAGGTATTTTGTCG-3′), *Il1r1* (F: 5′-CCCGAGGTCCAGTGGTATAA-3′; R: 5′-CTTCAGCCACATTCCTCACC-3′), and *GAPDH* (F: 5′-CGGCCGCATCTTCTTGTG-3′; R: 5′-CCGACCTTCACCATTTTGTCTAC-3′). 

### 4.10. Western Blotting

Collected cumulus cells and oocytes were lysed with 2× loading buffer as western blot samples. A 12–15% SDS gel was used to separate proteins and then transfer the protein to a nitrocellulose membrane. After being sealed in non-fat milk for 1 h at 37 °C, the membrane containing protein was incubated with antibodies of IL1A (1:300; sc-12741, Santa Cruz, Dallas, TX, USA), IL1B (1:500; ab234437, Abcam, Cambridge, UK), IL1R1 (1:300; sc-393998, Santa Cruz), REIA (1:500; WL01980, Wanleibio, Shenyang, China), NFKB1 (1:500; WL01866, Wanleibio), and tubulin (1:1000; ab6046, Abcam). After washing in TBST, the membrane was incubated at room temperature with a homologous secondary antibody for 2 h. ECL Plus (GE, Piscataway, NJ, USA) and the Tanon 3900 chemiluminescence imaging system (Tanon, Beijing, China) were used to color and observe protein bands, respectively.

### 4.11. Statistical Analysis

One-way ANOVA and Student’s *t*-test were conducted using GraphPad Prism 9.0 software. The post hoc test was executed using Tukey’s multiple comparison test. The data of at least three independent biological repetitions are expressed as mean ± SD. In the statistics confirming the aging time experiment, no significant difference is indicated by NS (*p* > 0.05), and a significant difference is shown by *** (*p* < 0.001). The specific *p*-values for other data were labeled in the figures.

## 5. Conclusions

In conclusion, our study revealed that oocytes with postovulatory aging undergo mitochondrial dysfunction, abnormal accumulation of ROS, and increased early apoptosis. The transcriptome sequencing results confirmed aging-related changes in oocytes and cumulus cells. Cumulus cells were confirmed to accelerate POA via IL1–IL1R1 interaction (Figure 7). Although this study was conducted on mice, it also provides a valuable resource for deeper examinations of POA in clinical research.

## Figures and Tables

**Figure 1 ijms-24-03530-f001:**
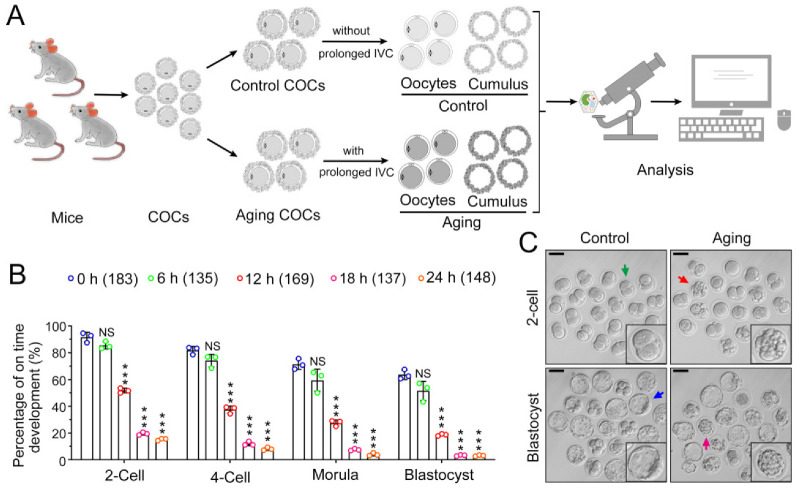
POA induces preimplantation embryo development failure. (**A**) Experimental design. The COCs were obtained from mice oviducts. All the COCs were collected for the pool and then were randomly selected and defined as the control group (without prolonged in vitro culture) or aging groups (with extended in vitro culture). The cumulus cells and oocytes were stripped after culture, and the two kinds of cells were separately collected for morphological detection, transcriptome analysis, and experiment verification. IVC: in vitro culture. (**B**) COCs underwent prolonged in vitro culture (0, 6, 12, 18, and 24 h) and IVF was performed. The percentages of fertilized oocytes developing into different stages were recorded. (**C**) Representative image of embryos at 24 h and 72 h after fertilization. Arrows indicate the embryo for enlargement. Green arrow indicates the normal 2-cells; red arrow indicates the degenerated embryos after fertilization; blue arrow indicates the developed blastocysts; pink arrow indicates the degenerated embryo. Scale bar = 100 μm. The data are presented as mean ± SD. A culturing duration of 0 h served as the control group; groups with extension culture for 6, 12, 18, and 24 h were compared with the control group, respectively. No significant difference is indicated by NS (*p* > 0.05) and a significant difference is shown by *** (*p* < 0.001). The number of oocytes is shown in brackets in the legend.

**Figure 2 ijms-24-03530-f002:**
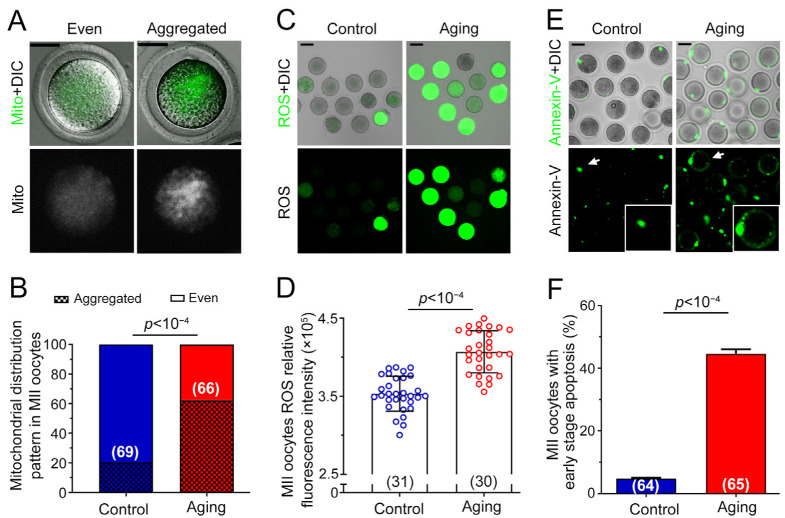
Abnormalities of oocytes were observed in POA. (**A**) Typical images of mitochondria distribution in oocytes. The green fluorescence represents mitotracker green FM combined specifically with mitochondria. Scale bar = 20 μm. (**B**) The proportion of oocyte mitochondria distribution patterns. (**C**) Representative images of ROS accumulation inside oocytes. The green fluorescence indicates ROS in oocytes detected using a DCFH-DA fluorescent probe. Scale bar = 100 μm. (**D**) Statistics of oocyte ROS fluorescence intensity. Each blue and red circle represents one detected oocyte in the control and aging groups, respectively. (**E**) Representative images of oocytes with early apoptosis. The white arrows indicate the embryos for enlargement. The green fluorescence Annexin-V on the oocyte cell membrane indicates early apoptosis. Scale bar = 20 μm. (**F**) Percentage of oocytes with early apoptosis. The data are presented as mean ± SD. *p* < 0.05 indicates a statistically significant difference. The number of oocytes is shown in brackets in the legend.

**Figure 3 ijms-24-03530-f003:**
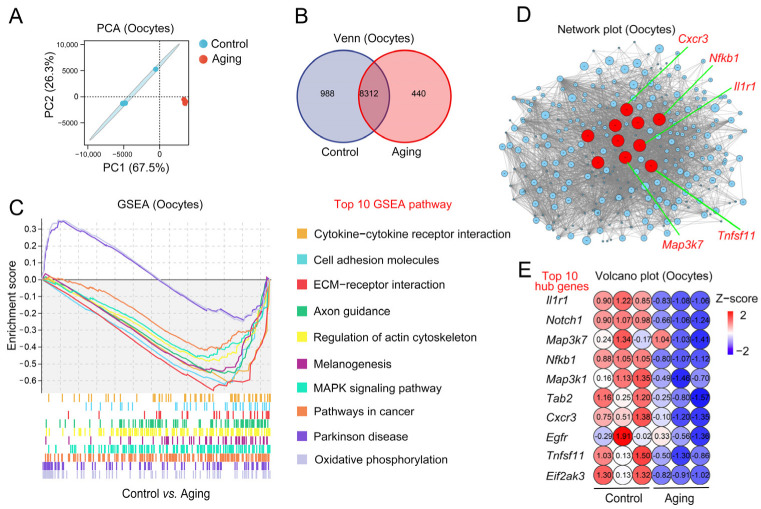
Changes to aging-related transcription levels in oocytes. (**A**) PCA diagram of oocytes’ Smart-seq 2. The blue circle stands for the control oocytes; the red circle stands for the aging oocytes. (**B**) Venn diagram of genes expressed in control oocytes and aging oocytes. (**C**) Representative diagram of top 10 aging-related GSEA pathways in control and aging oocytes. (**D**) Network plot exhibiting aging-related essential genes in oocytes. Each circle represents one individual gene; red circles indicate the top 10 hub genes. (**E**) Volcano plot showing the expression level of the top 10 aging-related hub genes in oocytes.

**Figure 4 ijms-24-03530-f004:**
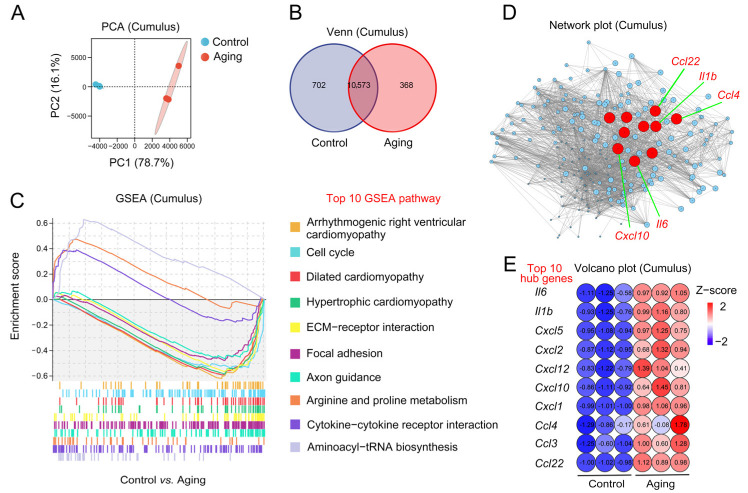
Changes to aging-related transcription levels in cumulus cells. (**A**) PCA diagram of cumulus cells’ RNA-seq. The blue circle stands for the control cumulus cells; the red circle stands for the aging cumulus cells. (**B**) Venn diagram of genes expressed in control cumulus cells and aging cumulus cells. (**C**) Representation diagram of top 10 aging-related GSEA pathways in control and aging cumulus cells. (**D**) The network plot exhibited aging-related essential genes in cumulus cells. Each circle represents one individual gene and the red circles indicate the top 10 hub genes. (**E**) The volcano plot shows the expression level of the top 10 aging-related hub genes in cumulus cells.

**Figure 5 ijms-24-03530-f005:**
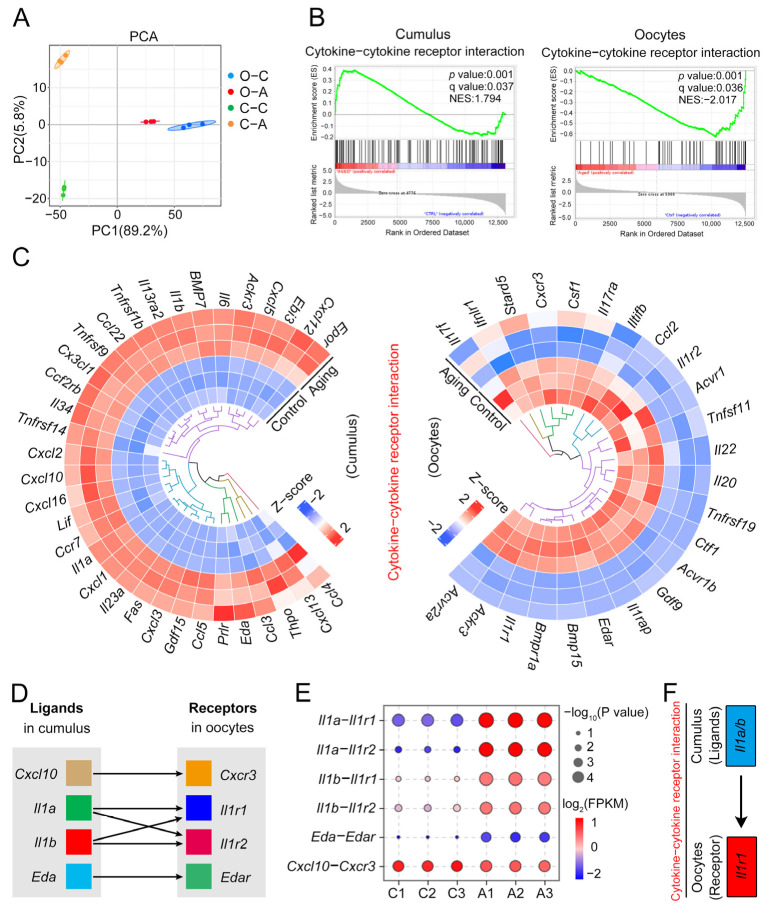
*Il1–Il1r1* participate in interactions between the cumulus cells and oocytes. (**A**) PCA diagram combined the sample of cumulus cells and oocytes. O–C: oocytes–control group; O–A: oocytes–aging group; C–C: cumulus cells–control group; C–A: cumulus cells–aging group. (**B**) GSEA enrichment plots of cytokine–cytokine receptor interaction in cumulus cells and oocytes. (**C**) Spin diagram of related genes in the cytokine–cytokine receptor interaction in cumulus cells (left) and oocytes (right). (**D**) Cumulus cell ligand genes (left) and oocyte receptor genes (right) are paired in the cytokine–cytokine receptor interaction. (**E**) Heatmap of ligand–receptor (*Il1a–Il1r1*; *Il1b–Il1r1*; *Il1a–Il1r2*; *Il1b–Il1r2*; *Eda–Edar*; *Cxcl10–Cxcr3*) expressed in control and aging groups. C: control group; A: aging group. (**F**) The representative diagram shows ligand *Il1* in cumulus cells and receptor *Il1r1* in oocytes paired in the cytokine–cytokine receptor interaction.

**Figure 6 ijms-24-03530-f006:**
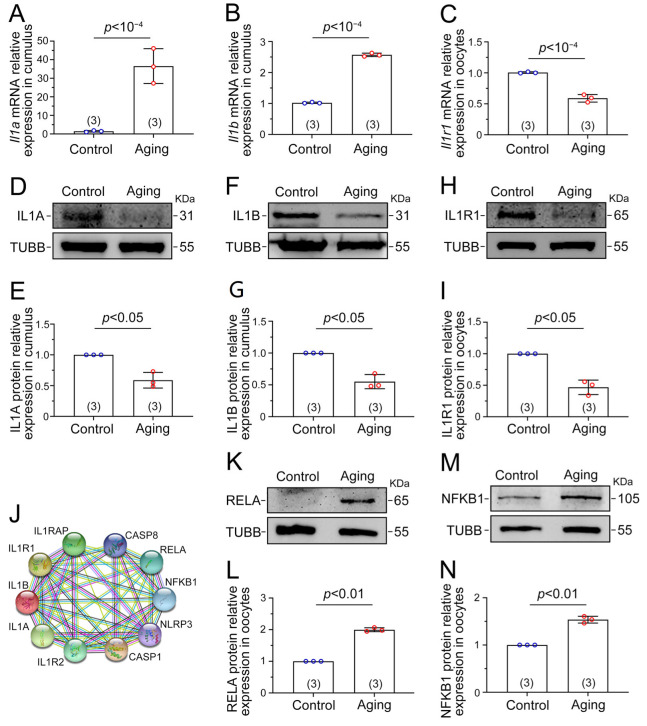
NF-κB signaling is activated by IL1–IL1R1 interaction. (**A**) Relative *Il1a* mRNA level in cumulus cells. (**B**) Relative *Il1b* mRNA level in cumulus cells. (**C**) Relative *Il1r1* mRNA level in oocytes. (**D**) The expression level of IL1A protein in cumulus cells. (**E**) The quantified intensity of IL1A/TUBB. (**F**) The expression level of IL1B protein in cumulus cells. (**G**) The quantified intensity of IL1B/TUBB. (**H**) The expression level of IL1R1 protein in oocytes. (**I**) The quantified intensity of IL1R1/TUBB. (**J**) The string diagram shows the protein–protein interaction network. (**K**) The expression level of RELA protein in oocytes. (**L**) The quantified intensity of RELA/TUBB. (**M**) The expression level of NFKB1 protein in oocytes. (**N**) The quantified intensity of NFKB1/TUBB. Each lane requires 200 oocytes or corresponding cumulus cells for the protein detection mentioned above. Expression of TUBB was applied as the interior control. The data are presented as mean ± SD. *p* < 0.05 indicates a statistically significant difference. The number of experiments is marked in brackets in the legend.

**Figure 7 ijms-24-03530-f007:**
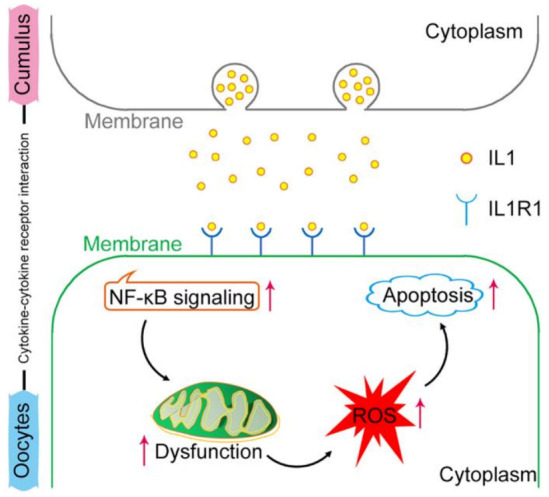
A schematic model showing that the cumulus cells accelerate POA through IL1–IL1R1 in cytokine–cytokine receptor interaction. This activates NF-κB signaling in oocytes, which in turn causes various abnormalities in oocytes, such as mitochondrial dysfunction, abnormal ROS accumulation, and early apoptosis, ultimately impairing oocyte quality.

## Data Availability

All data generated in this are available from the corresponding author upon request.

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
