# Peer review of "Cumulus Cells Accelerate Postovulatory Oocyte Aging through IL1–IL1R1 Interaction in Mice"

_ijms, 2023, doi:10.3390/ijms24043530_

Round 1
Reviewer 1 Report
Although the manuscript presented for review definitely does have a scientific value and provides novel, previously unpublished data, its main drawback is the absolutely unacceptable English. Numerous grammatical errors, wrong words and inappropriate for a research publication writing style altogether distract the reader from the essence of the study and render the manuscript impossible to read.
A thorough, careful and professional rewriting of the full paper is needed before it may be accepted for the next revision.
Due to significant difficulties in understanding of the study design and the need for a major revision, it makes no sense in commenting in detail on the text, although I have first started to do so for the Introduction part, and therefore include these comments below as well, but, as said, - the whole paper needs to be rewritten.
One of the main general comments: Experiment design is not very clear. An important question is: if the hypothesis was that cumulus cells might accelerate oocyte aging after ovulation, why was there no control group consisting of cumulus-free oocytes? Would the oocytes in the absence of cumulus cells preserve their quality in vitro longer?
Introduction:
L.16 and early apoptosis increased ⇒and increased early apoptosis
L.17, 29 decline of oocyte quality ⇒ decline in the oocyte quality
L.33 Cumulus cells feature promoting oocyte maturation ⇒ Cumulus cells promote oocyte maturation
P.42 NF-κB signaling plays essential roles ⇒ NF-κB signaling plays an essential role
L.47 Both can cause cell apoptosis, such as hyperandrogenism stimulates cumulus cells to secretion IL-1 to promote cell apoptosis in PCOS models --- not clear at all
L.49 At the same time, IL1 as ligands will bind to the cell surface receptors to regulate the body system -- what does it mean?
L. 50 Receptors include the signal-transmitting receptor IL1 receptor type 1 (IL1R1), and cannot transmit cell signals IL1 receptor type 2 (IL1R2) ⇒ IL-1 signal is transmitted through type 1 but not type 2 IL1 receptor
L.53 It is confirmed that preimplantation embryos may establish a relationship with maternal endometrium via IL1-IL1R1 interaction - either “confirmed” or “may”. So, I guess ‘It was shown that preimplantation embryos may establish…’
L.58 In this research, we will use transcriptomics analysis ⇒ In this study, we used transcriptomics analysis
L.59 - to reveal the reason by which cumulus cells accelerate POA ---? maybe to study the mechanism of POA acceleration by cumulus cells
Results:
Due to lots of grammatical and stylistic errors and multiple sentences with wrong tenses it was extremely hard to follow the story. But, in general:
Figure 1 C should be much larger to give the reader the chance to compare the defects
Stunted ⇒ arrested
What do the authors mean by ‘fragmented blastocyst’? It is not possible to see it from the image Fig 1C, but I assume that these are just degenerated, not fragmented, blastocysts there?
L.93 No significant difference is indicated by ND (P>0.05), and a significant difference is indicated by *** (P<0.001) ---- I see NS on the graph, not ND. Was the difference assessed in relation to the control group or what was compared vs what? Add this info.
L.94 The number of oocytes is shown in brackets. ⇒ The number of oocytes is shown in brackets in the legend
Figure 2E – the image is too small. Figure 2 C,E: in the legend no description of the markers used
Discussion:
more deep discussion of oocyte-cumulus cell interaction is needed.
Materials and Methods:
Additional data are needed: How many animals were used in total, how many COCs from one animal were recovered on average (mean ± SD). Were all oocytes randomly and equally distributed between the two groups, what about sperm donors – how many males in total used, how many were used for each IVF, were sperm samples pooled or not.
How many mice were used in the experiment in total? Age of the animals?
Experimental design is not really clear – were the oocytes pooled and then randomly divided into control and aging groups? How many males were used for IVF, was the sperm pooled?
IVF duration?
Author Response
Response to Reviewer 1:
- Numerous grammatical errors, wrong words and inappropriate for a research publication writing style altogether distract the reader from the essence of the study and render the manuscript impossible to read. A thorough, careful and professional rewriting of the full paper is needed before it may be accepted for the next revision.
Response: We have done a careful and professional rewriting of the full paper. We asked for language editing services to check the grammar, spelling, punctuation, and phrasing of the paper to improve its readability. Meanwhile, professionals are invited to check the overall structure, flow, and clarity of expression to suit the writing style of the research publication. We hope the revised manuscript could satisfy the reviewers.
- One of the main general comments: Experiment design is not very clear. An important question is: if the hypothesis was that cumulus cells might accelerate oocyte aging after ovulation, why was there no control group consisting of cumulus-free oocytes? Would the oocytes in the absence of cumulus cells preserve their quality in vitro longer?
Response: We improved the content of the experimental design in Figure 1A. A detailed description of the experimental design was added in the materials and methods section.
In the study, MII stage oocytes with cumulus cells (cumulus-oocyte complexes, COCs) were used to reveal the mechanism of POA. Our previous study showed that fertilization and early embryo development rates are extremely low when denuded oocytes (DOs) were used for fertilization. The fertilization rate of COCs was 92%, while that of DOs was only 68%. The 4-cell, morula and blastocyst generated from COCs were 80%, 75% and 53% respectively; while the 4-cell, morula and blastocyst obtained from DOs were 30%, 22% and 16%, respectively [1]. Another reason for not using DOs is that in the field of clinical and animal reproduction, COCs are usually used for fertilization. It has been proven that cumulus cells play an essential roles during fertilization for screening high-quality sperm. During IVF, interaction of sperm with cumulus cells facilitate fertilization and subsequent embryo development [2]. Thus, we don’t think samples of DO is suitable to reveal the POA mechanism.
We appreciate the reviewer’s insightful suggestion. We did not find any reports illustrating if oocytes in the absence of cumulus cells preserve their quality in vitro longer. However, it is an interesting topic and worth to be revealed by systematic study in the future.
Introduction:
- 16 and early apoptosis increased ⇒ and increased early apoptosis.
Response: We replaced it with “and increased early apoptosis”.
- 17, 29 decline of oocyte quality ⇒ decline in the oocyte quality.
Response: We replaced it with “decline in the oocyte quality”.
- 33 Cumulus cells feature promoting oocyte maturation ⇒ Cumulus cells promote oocyte maturation.
Response: We replaced it with “Cumulus cells promote oocyte maturation”.
- 42 NF-κB signaling plays essential roles ⇒ NF-κB signaling plays an essential role.
Response: We replaced it with “NF-κB signaling is essential in regulating cellular responses ”.
- 47 Both can cause cell apoptosis, such as hyperandrogenism stimulates cumulus cells to secretion IL1 to promote cell apoptosis in PCOS models --- not clear at all.
Response: The sentence was revised into“In the PCOS model, excessive hyperandrogenism stimulates cumulus cells secreting IL1A and IL1B, which in turn promotes oocytes apoptosis”.
- 49 At the same time, IL1 as ligands will bind to the cell surface receptors to regulate the body system -- what does it mean?
Response: The sentence was revised into “IL1A and IL1B are usually bind to the receptor on the cell surface and activate the downstream signal pathway”.
- 50 Receptors include the signal-transmitting receptor IL1 receptor type 1 (IL1R1), and cannot transmit cell signals IL1 receptor type 2 (IL1R2) ⇒ IL-1 signal is transmitted through type 1 but not type 2 IL1 receptor.
Response: We replaced it with “The IL1 signal is transmitted through the type 1 (IL1R1) but not the type 2 (IL1R2) IL1 receptor”.
- 53 It is confirmed that preimplantation embryos may establish a relationship with maternal endometrium via IL1-IL1R1 interaction - either ‘confirmed’ or ‘may’. So, I guess’It was shown that preimplantation embryos may establish….
Response: We replaced it with “It was shown that preimplantation embryos establish a relationship with maternal endometrium via IL1-IL1R1 interaction”.
- 58 In this research, we will use transcriptomics analysis ⇒ In this study, we used transcriptomics analysis.
Response: We replaced it with “In this study, we used transcriptomics analysis”.
- 59 - to reveal the reason by which cumulus cells accelerate POA ---? maybe to study the mechanism of POA acceleration by cumulus cells.
Response: We replaced it with “to study the mechanism of POA acceleration by cumulus cells”.
- Figure 1 C should be much larger to give the reader the chance to compare the defects.
Response: We enlarged Figure 1C and placed the detail of a typical 2-cell or blastocyst in the lower right corner.
- Stunted ⇒ arrested.
Response: We replaced it with “degenerated embryos”.
- What do the authors mean by ’fragmented blastocyst’? It is not possible to see it from the image Fig 1C, but I assume that these are just degenerated, not fragmented, blastocysts there?
Response: We replaced it with “degenerated embryos”.
- 93 No significant difference is indicated by ND (P>0.05), and a significant difference is indicated by *** (P<0.001) ---- I see NS on the graph, not ND. Was the difference assessed in relation to the control group or what was compared vs what? Add this info.
Response: We replaced it with “No significant difference is indicated by NS (P>0.05), and a significant difference is shown by *** (P<0.001)". In Figure 1B, the culturing duration of 0 hours served as the control group, groups with extension culture for 6, 12, 18 and 24 hours were compared with the control group, respectively. This information has been added to the revised manuscript.
- 94 The number of oocytes is shown in brackets. ⇒ The number of oocytes is shown in brackets in the legend.
Response: We replaced it with “The number of oocytes is shown in brackets in the legend”.
- Figure 2 E – the image is too small. Figure 2 C, E: in the legend no description of the markers used.
Response: We enlarged the typical oocyte in the lower right corner. In Figure 2 C, E the related description of markers was added.
- More deep discussion of oocyte-cumulus cell interaction is needed.
Response: We added a more profound discussion of oocyte-cumulus cell interaction in the discussion part of the revised manuscript in line 292-297.
- Additional data are needed: How many animals were used in total, how many COCs from one animal were recovered on average (mean ± SD). Were all oocytes randomly and equally distributed between the two groups, what about sperm donors–how many males in total used, how many were used for each IVF, were sperm samples pooled or not.
Response: We added this information in the materials and methods part. “This study used 200 ICR female mice (8 weeks old), including 15 mice in the transcriptome analysis and 185 mice in other experiments. In addition, three male ICR mice (16 weeks old) were used for sperm collection during IVF.
Female mice were killed by cervical dislocation, and COCs from the ampulla of the fallopian tube were collected. All the COCs were pooled and randomly divided into the control group (without prolonged in vitro culture) and the aging groups (with extended in vitro culture).
One sexually mature male mouse was used in each IVF repeat. A total of 3 repeats were conducted. All male ICR mice have been proven capable of fertilization through mating experiments. For sperm capacitation, male mice were killed by cervical dislocation. Sperm collected from the tail of both epididymides were mixed, and the sperm was suspended in 200 μL Tyrode (T6) medium supplemented with 10 mg/mL Bovine Serum Albumin (BSA) for 1.5 hours.”
- How many mice were used in the experiment in total? Age of the animals?
Response: This study used 200 ICR female mice (8 weeks old), including 15 mice in the transcriptome analysis and 185 mice in other experiments. In addition, three male ICR mice (16 weeks old) were used for sperm collection during IVF.
- Experimental design is not really clear–were the oocytes pooled and then randomly divided into control and aging groups? How many males were used for IVF, was the sperm pooled?
Response: We revised this part into “Female mice were killed by cervical dislocation, and COCs from the ampulla of the fallopian tube were collected. All the COCs were pooled and randomly divided into the control group (without prolonged in vitro culture) and the aging groups (with extended in vitro culture). The cumulus cells and oocytes were stripped after culture, and these two types of cells were separately collected for morphological detection, transcriptome analysis and other experiments (Figure 1A).”
Three male ICR mice (16 weeks old) were used for sperm collection during IVF. Sperm collected from the tail of both epididymides were mixed, and the sperm was suspended in 200 μL Tyrode (T6) medium supplemented with 10 mg/mL Bovine Serum Albumin (BSA) for 1.5 hours.
- IVF duration?
Response: IVF duration was stated in the revised manuscript.
References:
- Zhou CJ, Wu SN, Shen JP, Wang DH, Kong XW, Lu A, Li YJ, Zhou HX, Zhao YF, Liang CG: The beneficial effects of cumulus cells and oocyte-cumulus cell gap junctions depends on oocyte maturation and fertilization methods in mice. PeerJ 2016, 4:e1761.
- Xie L, Ma R, Han C, Su K, Zhang Q, Qiu T, Wang L, Huang G, Qiao J, Wang J, Cheng J: Integration of sperm motility and chemotaxis screening with a microchannel-based device. Clin Chem 2010, 56:1270-1278.
Reviewer 2 Report
The manuscript by Xin Web et al reported a large-scale study on possible involvement of IL1-IL1R1 interactions in cumulus cells and oocyte in regulation of molecular mechanism of oocyte ageing. Authors hypothesized about NFKB1 pathway involvement in this process.
Authors performed a large set of experiments including morphological and functional characterization of control and aged COC, oocyte and CC transcriptomics by RNAseq, differential analyses and GSEA, validation by Western blot. Results are well-illustrated and described. However, they are poorly discussed, although global transcriptome analysis of CC allowed to discover principle pathways involved in oocyte ageing process (4 figures are related to that!). These finding should be more discussed before the data on IL1 -IL1 receptor expression in CC and oocyte, respectively, which involvement and mechanism of action via NFKB1 are more speculative. Authors should be more accurate in their statements about possible mechanisms of actions.
English should be corrected.
L36-37 “most studies have confirmed that cumulus cells may promote oocytes aging and reduce the quality of oocytes” (17) - only one reference is given?
Does only secretion may explain that mRNA and protein levels of IL-proteins in CC are not correlated and varied in opposite directions between Control and Aged COCs?
Also, how does correlate a decreased level of ILR1 in aged oocytes to an increase of RELA and activation of NFKB1 pathway? Was such a mechanism already described in other cell models?
L300-301 is enzymatic degradation hypothesis? Should change to “may indicate they were …”
L303-305 Please give a reference or the data that synthesis of investigated proteins’ in MII oocytes did not occur if it is a hypothesis, it should be mentioned as that.
L235 Please give complete protein names of RELA and NFKB1 and explain the relation between these proteines
L325 what is DBTC1, please write complete name
Author Response
Response to Reviewer 2:
- They are poorly discussed, although global transcriptome analysis of CC allowed to discover principal pathways involved in oocyte ageing process (4 figures are related to that!). These finding should be more discussed before the data on IL1 -IL1 receptor expression in CC and oocyte, respectively, which involvement and mechanism of action via NFKB1 are more speculative. Authors should be more accurate in their statements about possible mechanisms of actions.
Response: In the revised manuscript, we added the relevant discussion on the transcriptional analysis of cumulus cells and oocytes, as well as the interaction of IL1-IL1R1 in cumulus cells and oocytes. The involvement of NFKB1 was obtained through the protein-protein interaction analysis network, and the mechanism was also verified by Western blot. According to the reviewer's suggestion, we have modified the manuscript to make the statement more accurate.
- English should be corrected.
Response: We invited a native English professional person checked the grammar, spelling, punctuation, and phrasing of the paper to improve its readability.
- L36-37 ‘most studies have confirmed that cumulus cells may promote oocytes aging and reduce the quality of oocytes’(17) - only one reference is given?
Response: We have added more references to support this description.
- Does only secretion may explain that mRNA and protein levels of IL-proteins in CC are not correlated and varied in opposite directions between Control and Aged COCs?
Response: We revised the corresponding part into “ The mRNA levels of ligands Il1a and Il1b were increased, and the protein levels were decreased in cumulus cells from the aging group. Therefore, we propose that cumulus cells would continuously synthesize mRNA after external stimulation during in vitro culture [1], as shown by increased mRNA levels of Il1a and Il1b, which generate more protein [2]. However, we found that the IL1A and IL1B protein expression levels decreased. It may be due to ligands protein being secreted extracellularly after enzymolysis [3, 4]. In addition, ubiquitin-proteasome is also a common mode of endogenous protein degradation [5]. For example, IL1A and IL1B proteins are polyubiquitinated and degraded by the proteasome in mouse bone marrow-derived dendritic cells [6]. Therefore, we speculate that the endogenous proteins IL1A and IL1B in cumulus cells may also undergo ubiquitin-proteasome action, leading to reduced protein expression. This explains that mRNA and protein expression levels of IL1A and IL1B are inconsistent in aging cumulus cells”.
- Also, how does correlate a decreased level of ILR1 in aged oocytes to an increase of RELA and activation of NFKB1 pathway? Was such a mechanism already described in other cell models?
Response: We revised the corresponding part into “IL1R1’s mRNA and protein expression are reduced in aging oocytes, because mammalian oocytes undergo mRNA decay, resulting in decreased mRNA expression [7, 8]. The receptor proteins are combined with ligands and are then internalized by cells. After internalization, some receptors are degraded or reused by the cell membrane, which can decrease the number of receptors protein [9, 10]. Although the expression of receptor IL1R1 protein decreased in oocytes, the downstream signaling pathway is activated after ligand-receptor binding. According to the protein-protein interaction network analysis, IL1-IL1R1 is related to RELA and NFKB1 proteins, which are the critical proteins in NF-κB signaling [11]. This study verified the increased expression of RELA and NFKB1 proteins in oocytes. Expression of RELA was increased in the liver cells in Il1r1 conditional deletion mice [12]. The results in the liver-injury model were consistent with those in this study, where decreased expression of IL1R1 protein was related to the increased expression of RELA protein. This suggests that the relationship between IL1R1 and RELA in somatic cells may also exist in germ cells. Although there is no direct evidence that the decrease in IL1R1 protein expression is related to the increase in NFKB1 protein expression, RELA usually forms heterodimers with NFKB1 to exert biological effects, and RELA protein expression is always positively correlated with NFKB1 protein expression [13, 14]. Therefore, the decrease in IL1R1 protein and the increase in RELA and NFKB1 protein expression found in this study are reasonable”.
- L300-301 is enzymatic degradation hypothesis? Should change to ‘may indicate they were…’
Response: We revised it into “It may be due to ligands protein being secreted extracellularly after enzymolysis”.
- L303-305 Please give a reference or the data that synthesis of investigated proteins’ in MII oocytes did not occur if it is a hypothesis, it should be mentioned as that.’
Response: We revised the corresponding part into “ IL1R1’s mRNA and protein expression are reduced in aging oocytes, because mammalian oocytes undergo mRNA decay, resulting in decreased mRNA expression [7, 8]. The receptor proteins are combined with ligands and are then internalized by cells. After internalization, some receptors are degraded or reused by the cell membrane, which can decrease the number of receptors protein [9, 10]”.
- L235 Please give complete protein names of RELA and NFKB1 and explain the relation between these proteines
Response: We added the complete protein names of RELA and NFKB1 and explained the relationship between these proteins. The supplementary contents are presented in the introduction part of revised manuscript.
- L325 what is DBTC1, please write complete name.
Response: We revised it into “dibutyltin dichloride”.
References:
- de los Santos MJ, Anderson DJ, Racowsky C, Simón C, Hill JA: Expression of interleukin-1 system genes in human gametes. Biol Reprod 1998, 59:1419-1424.
- Vorländer MK, Pacheco-Fiallos B, Plaschka C: Structural basis of mRNA maturation: Time to put it together. Curr Opin Struct Biol 2022, 75:102431.
- Frisch SM: Interleukin-1α: Novel functions in cell senescence and antiviral response. Cytokine 2022, 154:155875.
- Piccioli P, Rubartelli A: The secretion of IL-1β and options for release. Semin Immunol 2013, 25:425-429.
- Zhang T, Liu C, Li W, Kuang J, Qiu XY, Min L, Zhu L: Targeted protein degradation in mammalian cells: A promising avenue toward future. Comput Struct Biotechnol J 2022, 20:5477-5489.
- Ainscough JS, Frank Gerberick G, Zahedi-Nejad M, Lopez-Castejon G, Brough D, Kimber I, Dearman RJ: Dendritic cell IL-1α and IL-1β are polyubiquitinated and degraded by the proteasome. J Biol Chem 2014, 289:35582-35592.
- Brachova P, Alvarez NS, Christenson LK: Loss of Cnot6l Impairs Inosine RNA Modifications in Mouse Oocytes. Int J Mol Sci 2021, 22.
- Teruel M, Smith R, Catalano R: Growth factors and embryo development. Biocell 2000, 24:107-122.
- McNiven MA: Big gulps: specialized membrane domains for rapid receptor-mediated endocytosis. Trends Cell Biol 2006, 16:487-492.
- Kornilova ES: Receptor-mediated endocytosis and cytoskeleton. Biochemistry (Mosc) 2014, 79:865-878.
- Cartwright T, Perkins ND, C LW: NFKB1: a suppressor of inflammation, ageing and cancer. Febs j 2016, 283:1812-1822.
- Gehrke N, Hövelmeyer N, Waisman A, Straub BK, Weinmann-Menke J, Wörns MA, Galle PR, Schattenberg JM: Hepatocyte-specific deletion of IL1-RI attenuates liver injury by blocking IL-1 driven autoinflammation. J Hepatol 2018, 68:986-995.
- Catheline SE, Bell RD, Oluoch LS, James MN, Escalera-Rivera K, Maynard RD, Chang ME, Dean C, Botto E, Ketz JP, et al: IKKβ-NF-κB signaling in adult chondrocytes promotes the onset of age-related osteoarthritis in mice. Sci Signal 2021, 14:eabf3535.
- Rolova T, Dhungana H, Korhonen P, Valonen P, Kolosowska N, Konttinen H, Kanninen K, Tanila H, Malm T, Koistinaho J: Deletion of Nuclear Factor kappa B p50 Subunit Decreases Inflammatory Response and Mildly Protects Neurons from Transient Forebrain Ischemia-induced Damage. Aging Dis 2016, 7:450-465.
Round 2
Reviewer 1 Report
Please, find all comments integrated in the Manuscript as either required replacements of the text or as questions or suggestions to improve the content.

Author Response
Response to Reviewer 1:
- Change the wording for "detection experiment"
Response: We replaced it with “morphology, transcriptome, and other analysis”.
- Was the model generated? Then there should be a conclusion stating the best culture time to assess POA. Otherwise re-word the sentence to show that that oocyte aging was assessed over the culture period, at 6, 12, 18 and 24 h of culture.
Response: We added a conclusion “Therefore, we selected 12 hours as the suitable pre-culture time for setting up the model of POA”.
- There is one arrow of each color, so, change 'arrows' to 'arrow'.
Response: We replaced it with “arrow”.
- Oocyte numbers are hard to see on the bars in B and F.
Response: Our modification made the oocyte numbers in figures B and F clearer.
- Re-word this “ It was found that many molecules could be secreted from cumulus cells and play a role in oocytes”.
Response: We revised this part into “It has been found that some ions and small molecules can be transferred between cumulus cells and oocytes [1]”.
- ? re-word “the stable”.
Response: We revised this part into “This pathway is one of the key intercellular pathways that ensure normal cell development [2]”.
- transcriptomics or transcriptome analysis.
Response: We replaced it with “transcriptome analysis”.
- downstreat targets of NOX signalling?
Response: We replaced it with “activate NADPH oxidases”.
- Please, add some information to make this conclusion logical.
Response: We revised this part into “Therefore, the ligand-receptor interaction between the cumulus cells and oocytes plays an important role in accelerating POA. In addition to the Fas-FasL system, there may be other alternative ligand-receptor systems for cumulus cells to accelerate POA”.
- in aging/aged oocytes?
Response: We replaced it with “in aging oocytes”.
- is there a strong evidence? Or better write that a decreased IL1R1 expression was associated with an increased RELA expression?
Response: We replaced it with “a decreased IL1R1 expression was associated with an increased RELA expression”.
- Why does this substance come up here?
Response: We revised this part into “Similarly, mitochondrial dysfunction and elevated ROS levels have been observed after the IL1 activated NF-κB signaling in ataxic telangiectasia fibroblasts [3]”.
- Here I do not see a logical conlusion, add some more info and rewrite.
Response: We revised this part into “This is consistent with our results that cumulus cells interact with oocytes through the IL1-IL1R1 system in cytokine-cytokine receptor interactions, and activates NF-κB signaling in oocytes by increasing the expression of RELA and NFKB1, leading to mitochondrial dysfunction, abnormal ROS accumulation, and increased early apoptosis”.
- Control was used immediately, no incubation? Then state this.
Response: We revised this part into “All the COCs were pooled and incubated in a Chatot-Ziomek-Bavister (CZB) medium for 10 minutes. The oocytes were randomly divided into the control group and the aging group. Oocytes in the control group were immediately used for subsequent treatments without prolonged in vitro culture, while the oocytes in the aging group were subjected to extended in vitro culture before use”.
- in 4% PFA PBS?
Response: We replaced it with “PBS containing 4% (w/v) paraformaldehyde (PFA)”.
References
- Russell DL, Gilchrist RB, Brown HM, Thompson JG: Bidirectional communication between cumulus cells and the oocyte: Old hands and new players? Theriogenology 2016, 86:62-68.
- Qian Z, Zhang Z, Wang Y: T cell receptor signaling pathway and cytokine-cytokine receptor interaction affect the rehabilitation process after respiratory syncytial virus infection. PeerJ 2019, 7:e7089.
- Yoon J, Lee H, Lim JW, Kim H: Inhibitory effect of alpha-lipoic acid on mitochondrial dysfunction and interleukin-8 expression in interleukin-1beta-stimulated ataxia teleangiectasia fibroblasts. J Physiol Pharmacol 2020, 71.